Exercise interventions in migraine patients: a YouTube content analysis study based on grades of recommendation

Reina-Varona Álvaro 1 2
Rodríguez de Rivera-Romero Borja 2
Cabrera-López Carlos Donato 2
Fierro-Marrero José 2
Sánchez-Ruiz Irene 2
http://orcid.org/0000-0001-6379-6155 La Touche Roy 1 2 3 roylatouche@yahoo.es
1 Motion in Brains Research Group, Institute of Neuroscience and Sciences of the Movement (INCIMOV) , Aravaca, Madrid , Spain
2 Departamento de Fisioterapia, Centro Superior de Estudios Universitarios La Salle, Universidad Autónoma de Madrid , Aravaca, Madrid , Spain
3 Instituto de Dolor Craneofacial y Neuromusculoesquelético (INDCRAN) , Madrid, Madrid , Spain
Keogh Justin
Electronic publication date: 2022 Sep 30
Publication date: 2022
Volume: 10
Electronic Location ID: e14150
Received 2022 May 13; Accepted 2022 Sep 7
Copyright: © 2022 Reina-Varona et al.
Copyright year: 2022
Copyright holder: Reina-Varona et al.
License: This is an open access article distributed under the terms of the Creative Commons Attribution License, which permits unrestricted use, distribution, reproduction and adaptation in any medium and for any purpose provided that it is properly attributed. For attribution, the original author(s), title, publication source (PeerJ) and either DOI or URL of the article must be cited.
License URL: https://creativecommons.org/licenses/by/4.0/

Keywords: Migraine disorders, Exercise, Social network, YouTube, Content analysis

Funding: The authors received no funding for this work.

==============================
Background

Migraine is the second leading cause of disability worldwide, engendering a high economic cost in developed countries. The adverse events related to pharmacological treatment use have increased interest in non-pharmacological interventions such as exercise. YouTube offers a public source of information for migraine patients regarding exercise interventions for migraine improvement. However, this information has not been validated to ensure the quality and validity of its content.

Objective

This qualitative content analysis study aims to review and evaluate YouTube videos regarding exercise for migraine.

Methods

A systematic review of the current evidence regarding exercise for migraine was performed to establish evidence classification and grades of recommendation with the Scottish Intercollegiate Guidelines Network tool. The data sources were PubMed, PEDro, Cochrane, EBSCO, Google Scholar and Web of Science. The quality of YouTube videos on exercise in migraine was reviewed using the Global Quality Scale and DISCERN scale. Finally, the classification in grades of recommendation was used to evaluate the included videos in terms of the various exercise modalities.

Results

The classification into recommendation grades showed a grade B for aerobic exercise, yoga and changes in lifestyle behavior regarding exercise. A total of 129 videos were included. The healthcare and exercise professional authors produced higher quality videos with a significant statistical difference, although the alternative therapists and patients’ videos had a greater impact and a higher viewership based on correlation analysis. The evaluation of the videos based on the recommendation grades could only include 90 videos. 71% of these videos showed a B grade of recommendation, which corresponded to fewer than half of the total included videos.

Conclusions

YouTube needs higher quality videos on exercise for migraine, and healthcare and exercise professional authors need to improve their scoping strategies to acquire more views. The authors of YouTube videos should have better access to the best current evidence regarding exercise interventions in migraine.

Introduction

Migraine is a common health problem, with an estimated global age-standardized prevalence of 14.4%, the greatest impact of which is felt by women aged 15–49 years (Stovner et al., 2018). Migraine is defined as a recurrent headache disorder manifesting in attacks lasting 4–72 h, with characteristics of unilateral location, pulsating quality, moderate to severe intensity, aggravation by routine physical activity and association with nausea, photophobia and phonophobia (Olesen, 2018).

Migraine is the second leading cause of disability in the world, just after low back pain, with a major impact on work, family and society (Vos et al., 2017; Buse et al., 2019). Most patients perceive its negative impact on personal relationships, parenthood, careers, quality of life and health (Buse et al., 2019). Migraine’s impact on quality of life includes reduced physical activity, thereby resulting in high levels of sedentary behavior (Farris et al., 2018a). The disability generated by migraine incurs a high economic cost in developed countries, derived from healthcare evaluations, work absenteeism and low work efficiency (Berg & Stovner, 2005; Vos et al., 2017).

The primary treatment modality for migraine is pharmacological intervention based on the strongly recommended triptans and non-steroidal anti-inflammatory drugs for acute migraine management. For patients with chronic migraine, other pharmacological interventions, such as beta-blockers, valproic acid and topiramate, are recommended for migraine prophylaxis (Evers et al., 2009; Cameron et al., 2015; Ashina, 2020). Recently, the increased use of erenumab (a fully human monoclonal antibody) and botulinum toxin have had good results (Lattanzi et al., 2019; Herd et al., 2019; Ashina, 2020). However, none of these pharmacological interventions are free from adverse events (Evers et al., 2009; Lattanzi et al., 2019; Ashina, 2020).

There has therefore been increased interest in recent years in studying non-pharmacological interventions such as exercise to decrease the intake of medication and enhance the effects of both interventions in migraine symptoms. Aerobic exercise can reduce the intensity, frequency and duration of migraine episodes and improve the quality of life of patients with migraine (La Touche et al., 2020). Unlike pharmacological interventions, aerobic exercise does not provoke adverse events, although there is a widespread belief among patients with migraine that exercise can trigger a migraine episode, a belief associated with intentional avoidance of physical activity (Farris et al., 2018a).

Online health information can influence how patients deal with their health problems, and there is an increasing use of the Internet and social media to obtain health information. A study by AlMuammar et al. (2021) showed that 92.6% of the participants used the Internet to obtain medical information, 42% of them used this information because they did not want to go to the hospital, and YouTube was the second most frequently used social media for all searches. YouTube is one of the most ubiquitous forms of social media, with abundant information on various health topics. However, most of the sources that provide this information are not from healthcare professionals, influencing the videos’ quality. For example, only 9% of the authors of migraine videos on YouTube are healthcare professionals, and 44% of the non-healthcare practitioner videos showed complementary and alternative medicine as valid treatments for migraine (Saffi et al., 2020). The same could be true for videos on exercise interventions for migraine, which might not achieve sufficient quality and might hinder patients from exercising or could even generate adverse effects. With all this information, the main objective of the present study was to review and grade the current evidence regarding exercise in migraine. The second objective was to perform a content analysis of YouTube videos on exercise for migraine. The third objective was to evaluate the videos included in this study based on the grades of recommendation for exercise interventions for migraine.

Materials and Methods

Study design

We conducted a qualitative content analysis study with the following three phases: (1) a systematic review of the current evidence regarding exercise for migraine and a classification of the evidence and grades of recommendation; (2) a review of YouTube videos on exercise for migraine; (3) evaluation of the included videos based on grades of exercise and intervention recommendations for migraine.

In the first phase, the research group devised a search strategy to find the current evidence regarding exercise interventions for migraine, evaluated the quality of those selected studies and compiled a classification of evidence and grades of recommendation depending on the level of evidence of the included studies. In the second phase, the research group developed a search strategy to find YouTube videos on exercise for migraine, conduct an evaluation of their quality and determine their frequency data. In the third phase, the classification into grades of recommendation was used to evaluate the various reviewed videos according to their exercise modality and corresponding recommendation grade.

Ethical declarations

The videos included in this study are available on YouTube, which is a public domain platform. Therefore, no ethics approval was necessary for conducting the present study. The authors’ names of the included videos were kept anonymous in the present manuscript.

First phase: classification of evidence and grades of recommendation

Level of evidence and grades of recommendation

A systematic review of the current evidence regarding exercise prescription in migraine was performed in accordance with the Preferred Reporting Items for Systematic Reviews and Meta-analyses (PRISMA) using the harms checklist (Moher et al., 2009). This study was previously registered in PROSPERO, an international register for systematic reviews (PROSPERO, CRD42022316135). This literature review allowed us to create an evidence-based theoretical framework regarding the different exercise modalities and their efficacy.

The included studies and evidence classification were evaluated with the Scottish Intercollegiate Guidelines Network (SIGN), a tool for developing evidence-based clinical guidelines (Harbour & Miller, 2001). The advantages of this tool are the following: (1) levels of evidence are established attending to the study design and methodological quality of each included study; (2) guideline developers must take into account the generalizability, applicability, consistency and clinical impact of the evidence to link the evidence and recommendations; (3) grades of recommendation depend on the strength of the supporting evidence, which is based on the overall level of evidence and judgment of the guideline developers. The criteria for levels of evidence and grades of recommendation are shown in Table 1.

Table 1 SIGN levels of evidence and grades of recommendation.

Levels of evidence	Grades of recommendation	
1++	High-quality meta-analysis, systematic reviews of RCTs or RCTs with very low risk of bias	A	At least one meta-analysis, systematic review or RCT rated as 1++ and directly applicable to the target population OR	
1+	Well-conducted meta-analyses, systematic reviews of RCTs or RCTs with low risk of bias	A systematic review of RCTs or a body of evidence consisting principally of studies rated as 1+ directly applicable to the target population and demonstrating overall consistency of results	
1−	Meta-analyses, systematic reviews or RCTs, or RCTs with high risk of bias	B	A body of evidence including studies rated as 2++ directly applicable to the target population and demonstrating overall consistency of results OR	
2++	High-quality systematic reviews of case-control or cohort studies or
High-quality case-control or cohort studies with a very low risk of confounding, bias or chance and a high probability that the relationship is causal	Extrapolated evidence from studies rated as 1++ or 1+	
2+	Well-conducted case-control or cohort studies with a low risk of confounding, bias or chance and a moderate probability that the relationship is causal	C	A body of evidence including studies rated as 2+ directly applicable to the target population and demonstrating overall consistency of results OR	
2−	Case-control or cohort studies with a high risk of confounding, bias or chance and a significant risk that the relationship is not causal	Extrapolated evidence from studies rated as 2++	
3	Non-analytic studies, e.g., case reports, case series	D	Evidence level 3 or 4 OR	
4	Expert opinion		Extrapolated evidence from studies rated as 2+	
Note:

RCTs: randomized controlled trials.

Inclusion criteria

The inclusion criteria for selecting the articles for evidence classification and grades of recommendation were based on the PICO model (Population, Intervention, Comparator and Outcome measure) (Stone, 2002). No language restriction was applied.

Population

The participants included were those with migraine and older than 18 years. The migraine diagnosis must have been given by a neurologist, and both episodic and chronic migraine were included in the study.

Intervention and Controls

The intervention was exercise in its various modalities (e.g., aerobic, resistance, yoga), and no type of comparator was excluded due to the variety of studies included.

Outcomes

The outcome measures were pain intensity, frequency and/or duration, disability and quality of life. Pain intensity was assessed mainly with the Visual Analogue Scale or the Numeric Pain Rating Scale; migraine frequency was evaluated as days with migraine per month, and duration of migraine was considered as the number of hours per migraine attack. Disability and quality of life were measured with scales such as the Migraine Disability Assessment and Migraine-Specific Quality of Life. These instruments have been shown to be valid and reliable for evaluating the mentioned outcome measures (Stewart et al., 2001; Hjermstad et al., 2011; Bagley et al., 2012).

Study design

We selected systematic reviews, randomized controlled trials, cohort and case-control studies, case series, case reports and narrative reviews.

Search strategy for the classification of evidence and grades of recommendation

The search strategy was conducted on PubMed, PEDro, Cochrane, EBSCO, Web of Science and Google Scholar, and the last search was carried out in April 2022. The search strategy combined medical subject headings (MeSH) and non-MeSH terms by using a Boolean operator (OR and/or AND). The most relevant terms were “Migraine” and “Exercise”. Appendix 1 shows the database search strategy.

Appendix 1 Search strategy for included studies.

PubMed	
("Migraine Disorders"[MeSH Terms] OR "migraine"[All Fields]) AND ("Exercise"[MeSH Terms] OR "Exercise"[All Fields] OR "Exercise Therapy"[MeSH Terms] OR "exercise therap*"[All Fields])	
("Migraine Disorders"[MeSH Terms] OR "migraine"[All Fields]) AND "Yoga"[All Fields]	
("Migraine Disorders"[MeSH Terms] OR "migraine"[All Fields]) AND "Tai Chi"[All Fields]	
("Migraine Disorders"[MeSH Terms] OR "migraine"[All Fields]) AND ("Qigong"[All Fields] OR "KIKO"[All Fields])	
("Migraine Disorders"[MeSH Terms] OR "migraine"[All Fields]) AND ("resistance exercise"[All Fields] OR "resistance training"[All Fields] OR "strength exercise"[All Fields] OR "strength training"[All Fields])	
("Migraine Disorders"[MeSH Terms] OR "migraine"[All Fields]) AND ("relaxation"[All Fields] OR "breathing"[All Fields])	
("Migraine Disorders"[MeSH Terms] OR "migraine"[All Fields]) AND "lifestyle"[All Fields] AND "exercise"[All Fields]	
*Filters: Systematic Review; Meta-Analysis; Observational Study; Clinical Trial; Randomized Controlled Trial; Case Reports; Review; Adult: 19+ years; Humans.	
PEDro	
Abstract & Title: migraine exercise	
Abstract & Title: migraine yoga	
Abstract & Title: migraine/Therapy: fitness training	
Abstract & Title: migraine/Therapy: strength training	
Abstract & Title: migraine exercise/Therapy: behavior modification	
Cochrane	
(migraine) AND (exercise)	
(migraine) AND (yoga)	
(migraine) AND (tai chi)	
(migraine) AND ((qigong) OR (kiko))	
(migraine) AND ((resistance exercise) OR (strength exercise))	
(migraine) AND ((relaxation) OR (breathing))	
(migraine) AND (lifestyle) AND (exercise)	
EBSCO	
TI migraine AND exercise	
TI migraine AND yoga	
TI migraine AND Tai Chi	
TI migraine AND Qigong OR TI KIKO	
TI migraine AND TI resistance exercise OR TI strength exercise	
TI migraine AND TI relaxation techniques OR TI breathing exercises	
TI migraine AND TI lifestyle AND exercise	
*Filters: all adult: 19+ years	
Google Scholar	
allintitle: migraine aerobic OR exercise	
allintitle: migraine yoga	
allintitle: migraine Tai Chi	
allintitle: migraine Qigong OR KIKO	
allintitle: migraine resistance OR strength	
allintitle: migraine relaxation OR breathing	
allintitle: migraine lifestyle	
Web of Science	
(TS=(migraine)) AND TS=(exercise)	
(TS=(migraine)) AND TS=(aerobic exercise)	
(TS=(migraine)) AND TS=(Yoga)	
((TS=(Qigong)) OR TS=(KIKO)) AND TS=(migraine)	
((((TS=(resistance exercise)) OR TS=(resistance training)) OR TS=(strength exercise)) OR TS=(strength training)) AND TS=(migraine)	
((TS=(relaxation)) OR TS=(breathing)) AND TS=(migraine)	
((TS=(lifestyle)) AND TS=(exercise)) AND TS=(migraine)	
*Filters: Articles; Humans; Migraine Disorders	

Two independent reviewers (JFM and ISR) performed the search using the same methodology, and they resolved any differences that could emerge during this phase by consensus. Moreover, the reviewers screened the reference sections of the original studies manually. If necessary, the authors were contacted for detailed information.

Selection criteria and data extraction

The two independent reviewers assessed the relevance of the studies regarding the formulated questions and objectives and examined the study’s title information, abstract and keywords. If abstract information was insufficient or there was no consensus, the full text was revised. Then, the full text of the studies that met all the inclusion criteria was assessed. Differences that emerged during this phase were resolved by discussion with a third reviewer (Furlan et al., 2009).

Methodological quality assessment

Two independent reviewers evaluated the methodological quality of the included studies. For the quality assessment of the systematic reviews, the Modified Quality Assessment Scale for Systematic Reviews (AMSTAR) developed by Barton, Webster & Menz (2008) was employed. This scale has been shown to be valid and reliable for assessing the methodological quality of systematic reviews. It contains 13 items, each worth two points (“yes”, two points; “in part”, one point; “no”, 0 points), being the highest score 26. A score of 20 or more is considered to indicate high quality.

We used the PEDro scale to assess the methodological quality of the randomized controlled trials and quasi-experimental studies included in this guide. This scale has been shown to be a valid and reliable tool for the quality assessment of randomized controlled trials and assesses 11 items: (1) study eligibility criteria; (2) random allocation of participants; (3) concealed allocation; (4) comparability of groups at baseline; (5) participant blinding; (6) therapist blinding; (7) assessor blinding; (8) dropouts; (9) intention-to-treat analysis; (10) intergroup statistical comparison; and (11) point measures and variability data. Items are scored as yes (one point), no (0 points) or unknown (0 points), with a total score classified as (1) poor, 0–3 points, (2) fair, 4–5; (3) good, 6–8; or (4) excellent, 9–10 (de Morton, 2009).

We assessed the methodological quality of the selected cohort and case-control studies using the modified version of the Newcastle-Ottawa Quality Assessment Scale (NOS). This scale presents moderate inter-rater reliability and evaluates three criteria, based on nine questions, with a range of 0 to 4 stars: grade selection of participants; assessment of exposure, outcomes, and comparability; and control of confounding variables. The tallied stars provide four categories of study quality: (1) poor, 0–3 stars; (2) fair, 4–5 stars; (3) good, 6–7 stars; and (4) excellent, 8–9 stars (Wells et al., 2014).

The quality assessment of the case series included in this guide was assessed with the National Institutes of Health (NIH) Study Quality Assessment Tool for Case Series Studies. Each study’s summary score was calculated as a percentage with four scoring categories: (1) poor, 0–25%; (2) fair, 26–50%; (3) good, 51–75%; and (4) excellent, 76–100% (Kim, 2020).

The methodological quality of the case reports was evaluated with the Joanna Bridges Institute Critical Appraisal Checklist for Case Reports (JBI), which consists of eight questions with yes/no/unclear responses (Moola et al., 2020).

The methodological quality of the included narrative reviews was assessed with the Scale for the Assessment of Narrative Review Articles (SANRA), which contains six items, rated from 0 (low standard) to two (high standard), and covers the following topics: justification of the article’s importance for the readership; statement of the concrete aims or formulation of the questions; description of the literature search; referencing; scientific reasoning; and appropriate presentation of data. This scale has been shown to be valid and reliable, with sufficient internal consistency and inter-rater reliability (Baethge, Goldbeck-Wood & Mertens, 2019).

Risk of bias assessment

The risk of bias was evaluated with the Risk of Bias in Systematic Reviews Tool (ROBIS). This scale comprises three phases: (1) assessing relevance, which is optional; (2) identifying concerns about bias in the review process, which contains four domains to evaluate: study eligibility criteria; identification and selection of the studies; data collection and study appraisal; and synthesis and findings; (3) judging risk of bias. Each domain is responded to as “yes”, “probably yes”, “probably no”, “no” or “no information” to evaluate if they have been correctly addressed. In consequence, risk of bias is classified as “low”, “high” or “unclear” (Whiting et al., 2016).

The risk of bias of randomized controlled trials and quasi-experimental studies was assessed with the revised ROB 2.0 scale, which contains five domains: (1) bias arising from the randomization process; (2) bias due to deviations from intended interventions; (3) bias due to missing outcome data; (4) bias in measuring the outcome; and (5) bias in selecting the reported result. The risk of bias assessment items were classified as “high” if they presented a high risk of bias or “low” if they presented low risk of bias or “some concerns” (Sterne et al., 2019).

Second phase: YouTube video review

Search strategy for YouTube videos

We conducted a search on YouTube (https://www.youtube.com) between November 27, 2021 and January 25, 2022. The search strategy contained the terms “migraine exercise” in English and “migraña ejercicio” in Spanish. These terms were selected by the research group, which consisted of six physical therapists. Two independent reviewers (BRRR and CDCL) conducted two separate searches with the same methodology on their own computers through an incognito tab in their browser. Differences that emerged during this phase were resolved by consensus with the intervention of a third independent reviewer. Both search strategies were performed in two different phases. The first phase was conducted in November and December, and the second in January. No filter was used to replicate a simple search conducted by a patient.

Inclusion and exclusion criteria

The following inclusion criteria were employed: (1) videos on migraine and exercise and (2) videos in English or Spanish or whose subtitles were in one of these languages.

The following exclusion criteria were employed: (1) videos that did not mention migraine, (2) videos that did not include any type of exercise as a migraine intervention, (3) advertising videos and (4) non-educational videos.

Data collection

The following data were recorded for descriptive characteristics: number of likes, number of dislikes, number of views, upload date, days online, number of comments, duration, author, and video content. With this information, we calculated the impact and popularity of the videos with the Video Power Index (VPI) (VPI = like count/(like count + dislike count) × 100) and View Ratio (VR) (VR = view count/days online) (Erdem & Karaca, 2018; Kuru & Erken, 2020). To better access all the video data, we used the “Pafy” functional scripts from the YouTube data extraction project (https://github.com/phydiegoton/yt_data_extraction_project). To collect data on dislikes, we use the Return YouTube Dislike Chrome extension (https://chrome.google.com/webstore/detail/return-youtube-dislike/gebbhagfogifgggkldgodflihgfeippi) due to YouTube’s policy of concealing dislikes that went into effect on November 2021.

Coding of the various categories

To code the categories included in this study, we reviewed previous content analysis studies on health-related topics and developed tentative categories regarding the type of author and exercise modality (Lee et al., 2018; Kocyigit et al., 2019; Koçyiğit, Okyay & Akaltun, 2020; Koçyiğit, Akyol & Şahin, 2021; Chang & Park, 2021; Rodriguez-Rodriguez et al., 2021). This process was performed by consensus with all participants during the first meetings. After reviewing all the videos, these tentative categories were redefined by consensus, paying attention to all the possibilities observed in the various videos. Due to the necessity of limiting the number of categories for the statistical analysis, a number of the categories regarding exercise modality were grouped by similarity.

After this last consensus, the authors of the studies were classified into five separate categories: healthcare professionals, exercise professionals, alternative medicine therapists, non-healthcare professionals and patients. Moreover, we did another author classification based on the level of academic training: healthcare/exercise professional authors, which included healthcare professionals and exercise professionals, and alternative medicine/patient authors, which included alternative medicine therapists, non-healthcare professionals and patients. Exercise modalities were classified into nine separate categories: aerobic exercise, strength exercise, stretch/mobility, yoga/tai chi, meditation/breathing/relaxation, alternative therapies, self-massage/posture, vestibular rehabilitation and lifestyle/behavior changes regarding exercise.

Methodological quality assessment of the videos

The two independent reviewers assessed the methodological quality of the videos with two different scales. The Global Quality Scale (GQS) assesses the quality of videos based on information quality, flow quality and usefulness of the information for patients (Bernard et al., 2007). The score ranges from one to five points, with five points indicating the highest quality.

The modified 5-point DISCERN tool adapted from Charnock et al. (1999) assesses the quality and reliability of the information presented in the videos (Charnock et al., 1999; Singh, Singh & Singh, 2012). This scale was originally developed to assess the quality of information regarding treatment choices for health problems (Charnock et al., 1999) and includes five questions, each representing a separate quality criterion, with a total score ranging from one to five points (1, very poor; 2, poor; 3, average; 4, high; 5, very high).

The characteristics of both scales are shown in Table 2.

Table 2 Global Quality Scale for evaluating the information quality, flow quality and usefulness and 5-point DISCERN scale for evaluating the video quality and reliability.

GQS	Description	DISCERN	
1	Poor quality, poor flow of the video, most information missing, not at all useful for patients	-  Are the aims clear and achieved?	
2	Generally poor quality and poor flow, some information listed but many important topics missing, of very limited use to patients	-  Are reliable sources of information used? (e.g., publication cited, speaker is board-certified healthcare professional)	
3	Moderate quality, suboptimal flow, some important information is adequately discussed but others poorly discussed, somewhat useful for patients	-  Is the information presented balanced and unbiased?	
4	Good quality and generally good flow, most of the relevant information is listed, but some topics not covered, useful for patients	-  Are additional sources of information listed for patient reference?	
5	Excellent quality and flow, very useful for patients	-  Are areas of uncertainty mentioned?	

Third phase: video evaluation based on grades of recommendation

The included videos were rated with the various grades of recommendation depending on the exercise modality and its grading correspondence in the classification into grades of recommendation. We calculated the frequency of each grade of recommendation for the videos and compared the frequencies.

In this phase, the coding of the exercise modality was redefined due to the exclusion of those videos whose exercise categories were not supported by the classification into grades of recommendation. The new exercise modality obtained the following categories: aerobic exercise, resistance exercise, neck strengthening, breathing/relaxation exercise, yoga, Qigong, Tai Chi and lifestyle/behavior changes regarding exercise. Another four subcategories were included within aerobic exercise: high-intensity interval aerobic training; moderate-intensity aerobic training; high-intensity interval aerobic training/moderate-intensity aerobic training; and low-moderate continuous aerobic training.

Statistical analysis

To analyze the data obtained in this study, we used the Statistical Package for Social Sciences (SPSS) version 27.0 (IBM Inc., Chicago, IL, USA). Data normality was evaluated with the Kolmogorov–Smirnov test, and the variables are expressed as mean ± standard deviation, numbers and percentages. The associations between the quantitative and semiquantitative data were examined with Spearman’s correlation coefficient, which has various cut-offs: 0.00–0.10 indicates a negligible correlation; 0.10–0.39 indicates a weak correlation; 0.40–0.69 indicates a moderate correlation; 0.70–0.89 indicates a strong correlation; and 0.90–1.00 indicates a very strong correlation (Schober, Boer & Schwarte, 2018). The data were dichotomized into two categories: healthcare/exercise professional authors and alternative medicine/patient authors. This decision was made considering that healthcare and exercise professionals have a university education based on recognized evidence on the pathogenesis of migraine and/or in the exercise field for migraine treatment through exercise interventions (health professionals and exercise professionals). Those professionals whose interventions had no proven evidence in the migraine approach (alternative therapists) and those non-health professionals (yoga/Tai Chi instructors) whose education had not achieved the requirements necessary to apply an exercise intervention to a population with medical conditions were included in the alternative medicine/patient authors category along with the patients. The proportional difference between these two categories regarding the scores from the two quality scales and the assigned grades of recommendation were analyzed with the chi-squared test (Donner & Robert Li, 1990). We employed the Mann–Whitney U test to compare all quantitative and semi-quantitative data between the healthcare/exercise professional and alternative medicine/patient categories (Nachar, 2008). The statistical significance level was p < 0.05.

We calculated the Kappa coefficient (κ) and percentage of agreement between the reviewers’ scores to assess the reliability of all the quality and risk-of-bias scales prior to any consensus. The inter-rater reliability was estimated with κ: a value of κ < 0.5 showed a low level of agreement; a value of 0.5–0.7 showed a moderate level of agreement; and κ > 0.7 showed a high level of agreement (McHugh, 2012). The intervention of a third independent reviewer was necessary to reach a consensus on the final score if both independent reviewers disagreed.

Results

Classification of evidence and grades of recommendation

A total of 69 articles were included in the development of evidence classification and grades of recommendation: 36 clinical trials, seven systematic reviews, five cohorts, one case series, two case reports and 18 narrative reviews. Aerobic exercise, yoga and lifestyle/behavior changes regarding exercise obtained a B grade. The results with the various grades of recommendation regarding exercise for migraine are shown in Table 3. The kappa coefficient indicating agreement between the examiners was moderate for the evaluation of AMSTAR (κ = 0.512), ROBIS (κ = 0.588) and SANRA (κ = 0.681) and high for the PEDro scale (κ = 0.860), the ROB 2.0 scale (κ = 0.801), NOS (κ = 0.762) and NIH and JBI scales (κ = 1.000).

Table 3 Grades of recommendation of the various exercise modalities for migraine treatment.

Intervention	Study types	Grade of recommendation	Methodological quality	Risk of bias	YouTube (n = 129)	
Aerobic exercise	MMA: (Herranz-Gómez et al., 2021)	B
In favor of intervention	–	–	n = 24	
MA and SR of RCTs and q-RCTs: (La Touche et al., 2020), (Varangot-Reille et al., 2021), (Lemmens et al., 2019)	AMSTAR: 25/26
“High Quality”	ROBIS: “High Risk of Bias” to
“Low Risk of Bias”		
RCTs: (Oliveira et al., 2019), (Varkey et al., 2011), (Oliveira et al., 2017), (Pairo et al., 2016), (Ahmadi, 2015), (Eslami et al., 2021), (Matin, Taghian & Chitsaz, 2022), (Santiago et al., 2014), (Hanssen et al., 2018), (Hanssen et al., 2017)
q-RCTs: (Lafave, 1994), (Darabaneanu et al., 2011), (Lockett & Campbell, 1992), (Luedtke et al., 2020), (Overath et al., 2014), (Abdi, Parnow & Azizi, 2014), (Köseoglu et al., 2003), (Varkey et al., 2009).	PEDro:
-RCT: 4.4/10
-q-RCT: 2.75/10
“Poor” to “Good” quality	ROB 2.0: “High Risk of Bias”		
Cohort: (Hagan et al., 2021)	NOS: 6/9 “Good”	–		
Narrative Reviews: (Busch & Gaul, 2008b), (Amin et al., 2018), (Barber & Pace, 2020)
(Busch & Gaul, 2008a), (Song & Chu, 2021), (Patel & Minen, 2019), (Wells, Beuthin & Granetzke, 2019), (Irby et al., 2016), (Lippi, Mattiuzzi & Sanchis-Gomar, 2018), (Robblee & Starling, 2019), (Tepper, 2015), (Guarín-Duque et al., 2021), (Daenen et al., 2015), (Mauskop, 2012), (Hindiyeh, Krusz & Cowan, 2013)	SANRA: 8.46/12	–		
Yoga	SR of RCTs: (Wu et al., 2022)	B
In favor of intervention	AMSTAR: 17/26 “Low Quality”	ROBIS: “Low Risk of Bias”	n = 25	
RCTs: (Mehta et al., 2021), (John et al., 2007), (Kumar et al., 2020), (Sathyaprabha et al., 2014), (Mohammadi et al., 2020), (Boroujeni et al., 2015)		PEDro:
-RCT: 4.6/10
“Poor” to “Good” quality	ROB 2.0: “High Risk of Bias” to “Some concerns”		
Narrative Reviews: (Amin et al., 2018), (Barber & Pace, 2020), (Song & Chu, 2021), (Patel & Minen, 2019), (Wells, Beuthin & Granetzke, 2019), (Guarín-Duque et al., 2021)		SANRA: 9.33/12			
Multimodal treatment combinations: exercise (aerobic and/or resistance exercise), stretching, relaxation and/or psychological therapy	MA and SR of RCTs and q-RCTs: (Luedtke et al., 2016), (Brønfort et al., 2004)	B
In favor of intervention	AMSTAR: 21.5/26 “High Quality”	ROBIS: “Low Risk of Bias”		
RCT: (Mehta et al., 2021), (Dittrich et al., 2008)		PEDro: 6.5/10 “Good”	ROB 2.0: “Some Concerns”		
Cohort: (Gaul et al., 2011)		NOS: 3/9 “Poor”			
Narrative Review: (Song & Chu, 2021), (Lippi, Mattiuzzi & Sanchis-Gomar, 2018), (Becker & Sauro, 2009)		SANRA: 7/12			
Moderate intensity continuous aerobic exercise	RCTs: (Oliveira et al., 2019), (Varkey et al., 2011), (Oliveira et al., 2017), (Pairo et al., 2016), (Ahmadi, 2015), (Eslami et al., 2021), (Hanssen et al., 2018), (Hanssen et al., 2017)
q-RCTs: (Darabaneanu et al., 2011), (Luedtke et al., 2020), (Overath et al., 2014), (Varkey et al., 2009)	B
In favor of intervention	PEDro:
-RCT: 4.5/10
-q-RCT: 2.75/10
“Poor” to “Good” quality	ROB 2.0: “High Risk of Bias”	n = 9	
Narrative Reviews: (Busch & Gaul, 2008b), (Amin et al., 2018), (Barber & Pace, 2020), (Busch & Gaul, 2008a), (Song & Chu, 2021), (Patel & Minen, 2019), (Irby et al., 2016), (Lippi, Mattiuzzi & Sanchis-Gomar, 2018), (Robblee & Starling, 2019), (Tepper, 2015), (Guarín-Duque et al., 2021), (Daenen et al., 2015), (Mauskop, 2012), (Hindiyeh, Krusz & Cowan, 2013)		SANRA: 8.42/12			
Lifestyle/behavior changes regarding exercise intervention: aerobic exercise, diet, behavioral strategies	RCT: (Bond et al., 2018)	B
In favor of intervention	PEDro: 6/10 “Good”	ROB 2.0: “High Risk of Bias”	n = 18	
Cohort: (Seok, Cho & Chung, 2006), (Woldeamanuel & Cowan, 2016)		NOS: 7.5/9 “Good”			
Narrative Review: (Song & Chu, 2021), (Robblee & Starling, 2019), (Mauskop, 2012)		SANRA: 8/12			
High-intensity continuous aerobic exercise	RCTs: (Eslami et al., 2021), (Parnow et al., 2016)
q-RCTs: (Lockett & Campbell, 1992), (Abdi, Parnow & Azizi, 2014)	C
In favor of intervention	PEDro:
-RCT: 4/10
-q-RCT: 2.5/10
“Poor” to “Fair” quality	ROB 2.0: “High Risk of Bias”		
	Narrative Review: (Busch & Gaul, 2008b), (Busch & Gaul, 2008a), (Hindiyeh, Krusz & Cowan, 2013)		SANRA: 9/12			
High-intensity interval aerobic exercise	RCTs: (Matin, Taghian & Chitsaz, 2022), (Hanssen et al., 2018), (Hanssen et al., 2017)	C
In favor of intervention	PEDro:
RCTs: 3.33/10
“Poor” to “Fair” quality	ROB 2.0: “High Risk of Bias”	n = 2	
Narrative Reviews: (Barber & Pace, 2020), (Song & Chu, 2021), (Lippi, Mattiuzzi & Sanchis-Gomar, 2018)		SANRA: 9.33/12			
Relaxation intervention: relaxation breathing and/or progressive muscle relaxation or relaxation through biofeedback of physiological variables or mindfulness/meditation	RCTs: (Varkey et al., 2011), (Minen et al., 2020), (Meyer et al., 2016)	C
In favor of intervention	PEDro: 4.3/10
“Fair” to “Good” quality	ROB 2.0: “High Risk of Bias”	n = 14	
Narrative Review: (Busch & Gaul, 2008b), (Amin et al., 2018), (Barber & Pace, 2020), (Song & Chu, 2021), (Patel & Minen, 2019), (Wells, Beuthin & Granetzke, 2019), (Irby et al., 2016), (Meyer et al., 2018), (Robblee & Starling, 2019), (Tepper, 2015), (Guarín-Duque et al., 2021), (Mauskop, 2012), (Ahn, 2013), (Hindiyeh, Krusz & Cowan, 2013)		SANRA: 8.14/12			
Multimodal treatment combinations: manual therapy, exercise (aerobic and/or resistance exercise) and/or behavioral strategies	RCT: (Lemstra, Stewart & Olszynski, 2002)	C
In favor of intervention	PEDro: 7/10 “Good”	ROB 2.0: “Some Concerns”		
Case Reports: (Davis, 2003), (Suso-Martí, Swann & Cuenca-Martínez, 2019)		–	JBI: “Moderate Risk of Bias”		
Narrative Review: (Busch & Gaul, 2008b), (Becker & Sauro, 2009)		SANRA: 6.5/12			
Diaphragm respiratory training, cervical mobilization and traction, massotherapy, digital compression trigger points and passive stretching of neck muscles	RCT: (Bevilaqua-Grossi et al., 2016)	C
In favor of intervention	PEDro: 6/10 “Good”	ROB 2.0: “Some Concerns”		
Neck strength exercises	ECA: (Benatto et al., 2022)	C
Against intervention	PEDro: 6/10 “Good”	ROB 2.0: “Some Concerns”		
Low-moderate continuous intensity aerobic exercise and amitriptyline	RCT: (Santiago et al., 2014)	C
In favor of intervention	PEDro: 4/10 “Fair”	ROB 2.0: “High Risk of Bias”		
Narrative Reviews: (Amin et al., 2018), (Barber & Pace, 2020), (Song & Chu, 2021), (Robblee & Starling, 2019)		SANRA: 9.75/12			
Aerobic, resistance and postural exercises program (cycling, walking, stepper, training upper extremities, neck postural exercises, rowing) and medication	q-RCT: (Narin et al., 2003)	C
In favor of intervention	PEDro: 5/10 “Fair”	ROB 2.0: “Some Concerns”		
Narrative (Busch & Gaul, 2008b), (Busch & Gaul, 2008a), (Song & Chu, 2021), (Irby et al., 2016), (Lippi, Mattiuzzi & Sanchis-Gomar, 2018), (Hindiyeh, Krusz & Cowan, 2013)		SANRA: 8.83/12			
Resistance exercise	RCT: (Aslani et al., 2021)	C
In favor of intervention	PEDro: 4/10 “Fair”	ROB 2.0: “High Risk of Bias”	n = 6	
Narrative Reviews: (Song & Chu, 2021), (Mauskop, 2012)		SANRA: 8/12			
High-intensity interval aerobic exercise, B12 and magnesium supplementation	RCT: (Matin, Taghian & Chitsaz, 2022)	C
In favor of intervention	PEDro: 4/10 “Fair”	ROB 2.0: “High Risk of Bias”		
Low-moderate continuous intensity Aerobic exercise and drug therapy	q-RCT: (Karimi et al., 2015)	C
In favor of intervention	PEDro: 5/10 “Fair”	ROB 2.0: “High Risk of Bias”		
Moderate-intensity continuous aerobic exercise and progressive muscle relaxation	q-RCT: (Maryum Naseer Butt et al., 2022)	C
In favor of intervention	PEDro: 5/10 “Fair”	ROB 2.0: “High Risk of Bias”		
Moderate–high-intensity aerobic exercise	q-RCT: (Lafave, 1994)	C
In favor of intervention	PEDro: 4/10 “Fair”	ROB 2.0: “High Risk of Bias”		
Cohort: (Hagan et al., 2021)		NOS: 6/9 “Good”			
Narrative Review: (Guarín-Duque et al., 2021)		SANRA: 7/12			
Low-intensity aerobic exercise	q-RCT: (Köseoglu et al., 2003)	C
In favor of intervention	PEDro: 2/10 “Poor”	ROB 2.0: “High Risk of Bias”		
Narrative Review: (Busch & Gaul, 2008b), (Amin et al., 2018), (Barber & Pace, 2020), (Busch & Gaul, 2008a), (Song & Chu, 2021), (Hindiyeh, Krusz & Cowan, 2013)		SANRA: 8.83/12			
Daily physical activity participation	Cohort: (Farris et al., 2018b)	D
In favor of intervention	NOS: 2/9 “Poor”			
Qigong	Case Series: (Elinoff et al., 2009)	D
In favor of intervention	NHI: “Excellent”		n = 2	
Narrative Reviews: (Amin et al., 2018), (Guarín-Duque et al., 2021)		SANRA: 9/12			
Tai Chi	Narrative Reviews: (Song & Chu, 2021), (Wells, Beuthin & Granetzke, 2019)	D
In favor of intervention	SANRA: 9,5/12		n = 1	
Quantitative warm-up	Narrative Review: (Hindiyeh, Krusz & Cowan, 2013)	D
In favor of intervention	SANRA: 5/12			
Note:

MA, Meta-Analysis; MMA, Meta-Meta-Analysis; q-RCTs, quasi-Randomized-Controlled Trials; RCTs, Randomized-Controlled Trials; SR, Systematic Review.

Selected videos from YouTube

Initially, 367 videos were selected for possible inclusion in this study. After going through the eligibility process (Fig. 1), 129 videos were ultimately included for the qualitative analysis. The kappa coefficient indicating agreement between the examiners in the evaluation of the GQS and DISCERN scales was high for both (κ = 0.831 and κ = 0.735, respectively).

Figure 1 Adapted video selection flowchart based on PRISMA.

The frequency data of the videos by GQS and DISCERN scores, author and subject are shown in Table 4. The descriptive statistics obtained from the videos are presented in Table 5, with their median and quartiles due to the data’s non-normal distribution.

Table 4 Distribution of videos by GQS and DISCERN scores, author and subject categories.

		No.	%	
GQS	Poor quality	92	71.3	
Generally poor quality	19	14.7	
Moderate quality	13	10.1	
Good quality	4	3.1	
Excellent quality	1	0.8	
Total	129	100	
DISCERN	Very poor	100	77.5	
Poor	15	11.6	
Average	9	7	
High	5	3.9	
Very high	0	0	
Total	129	100	
Author	Healthcare professional	38	29.5	
Exercise professional	15	11.6	
Alternative therapist	30	23.3	
Non-healthcare	43	33.3	
Patient	3	2.3	
Total	129	100	
Subject	Aerobic exercise	24	18.6	
Strength exercise	6	4.7	
Stretching/Mobility	19	14.7	
Yoga/Tai Chi	26	20.2	
Meditation/Breathing/Relaxation	14	10.9	
Alternative therapies	9	7	
Self-massage/Posture	7	5.4	
Lifestyle changes regarding exercise	18	14	
Vestibular rehabilitation	6	4.7	
Total	129	100	

Table 5 Descriptive statistics of the video characteristics in median and quartiles.

	Median	Quartile 1	Quartile 3	
Likes	64	4.5	816.5	
Dislikes	0	0	17	
Video power index	99.42%	94.88%	100%	
Visits	2,679	187	27,203.5	
Days online	827	454.5	1,702	
View ratio	3.4914	0.2794	42.1551	
Comments	8	0.75	56.75	
Duration	399	211.5	786.5	

The distribution of the GQS and DISCERN scores for the two author categories (healthcare/exercise professional authors and alternative medicine/patient authors) is shown in Table 6. The chi-squared test showed a statistically significant difference between these two categories (p < 0.001), with healthcare/exercise professional authors showing a larger percentage of high scores than alternative medicine/patient authors.

Table 6 Distribution of GQS and DISCERN scores within healtcare/exercise professional authors and alternative medicine/patient authors categories.

	Healthcare/exercise professional authors	Alternative medicine/patient authors	
GQS	Poor quality	25 (47.2%)	67 (88.2%)	
Generally poor quality	12 (22.6%)	7 (9.2%)	
Moderate quality	11 (20.8%)	2 (2.6%)	
Good quality	4 (7.5%)	0 (0%)	
Excellent quality	1 (1.9%)	0 (0%)	
Total	53 (100%)	76 (100%)	
DISCERN	Very poor	29 (54.7%)	71 (93.4%)	
Poor	10 (18.9%)	5 (6.6%)	
Average	9 (17%)	0 (0%)	
High	5 (9.4%)	0 (0%)	
Very high	0 (0%)	0 (0%)	
Total	53 (100%)	76 (100%)	

The Mann–Whitney U test also showed that the alternative medicine/patient authors’ videos obtained more likes, dislikes, visits, visits per day (VR) and comments than the healthcare/exercise professional videos, with statistically significant differences (p < 0.05). In contrast, there were no differences in VPI, days online or duration between groups (Table 7).

Table 7 Differences between healthcare/exercise professionals authors and alternative medicine/patient authors in the quantitative characteristics of the videos.

		Median	Q1	Q3	P	
Likes	Healthcare/exercise	21	2	138	0.002	
Alternative/patient	129.5	10.5	1,116	
Dislikes	Healthcare/exercise	0	0	4	0.007	
Alternative/patient	4	0	39.75	
VPI	Healthcare/exercise	100%	94.77%	100%	0.262	
Alternative/patient	98.75%	95.06%	100%	
Visits	Healthcare/exercise	892	83.5	7,577.5	0.004	
Alternative/patient	8,269	239.5	48,983.5	
Days online	Healthcare/exercise	978	470	1839	0.352	
Alternative/patient	717	406.75	1,544.25	
VR	Healthcare/exercise	1.24	0.11	10.26	0.002	
Alternative/patient	7.46	0.53	77.71	
Comments	Healthcare/exercise	2.5	0	20.5	0.008	
Alternative/patient	15.5	1.75	115.25	
Duration	Healthcare/exercise	305	149.5	726	0.138	
Alternative/patient	432.5	237.5	835.25	
Note:

Healthcare/exercise, healthcare/exercise professional authors; Alternative/patient, alternative medicine/patient authors; VPI, Video Power Index; VR, view ratio.

The correlation analysis showed strong correlations between GQS and DISCERN. There were statistically significant negative weak correlations between GQS and likes, dislikes, visits, VR and comments. Moreover, there were statistically significant negative weak correlations between DISCERN and likes, dislikes, visits, VR and comments (Table 8).

Table 8 Correlation analysis of the quality scales and the characteristics of the videos.

	GQS	DISCERN	Likes	Dislikes	VPI	Visits	Days online	VR	Comments	
DISCERN	0.89**									
Likes	−0.39**	−0.35**								
Dislikes	−0.26**	−0.20*	0.86**							
VPI	−0.04	−0.12	−0.24**	−0.59**						
Visits	−0.37**	−0.33**	0.95**	0.88**	−0.36**					
Days online	0.04	−0.03	0.10	0.18*	−0.22*	0.30**				
VR	−0.40**	−0.32**	0.95**	0.85**	−0.30**	0.94**	−0.02			
Comments	−0.35**	−0.31**	0.92**	0.80**	−0.32**	0.90**	0.96	0.91**		
Duration	−0.16	−0.14	0.29**	0.19*	0.03	0.21*	−0.23**	0.29**	0.28**	
Notes:

VPI, Video Power Index; VR, view ratio.

* P < 0.05.

** P < 0.001.

Evaluation of the videos based on grades of recommendation

The evaluation of the videos regarding the grades of recommendation based on SIGN included 90 of the 129 videos. The videos excluded from the SIGN evaluation showed interventions that have been discarded from the classification into grades of recommendation due to lack of evidence.

The evaluation of the included videos showed a B grade of recommendation for those videos about general advice on aerobic exercise or moderate-intensity aerobic exercise, yoga and lifestyle/behavior changes regarding exercise. These videos accounted for approximately 71% of all videos included in the SIGN evaluation and close to 50% of the total videos included in the current study. The percentage of videos regarding their interventions and grades of recommendation are shown in Fig. 2.

Figure 2 Percentage of videos regarding grades of recommendation.

IFT, In favor of intervention; AT, Against the intervention.

Regarding the distribution of the grades of recommendation of the videos based on their authors’ category, there was no statistically significant difference between both categories (p = 0.198). This distribution is shown in Fig 3.

Figure 3 Percentage of videos based on grades of recommendation and author categories.

Discussion

The purpose of this study was to evaluate the quality of YouTube videos regarding prescribed exercise and advice for patients with migraine. We also evaluated the evidence levels and grades of recommendation for prescribed exercise for migraine, which made it possible to assess the quality of the videos based on objective and evidence-based criteria about modalities and parameters regarding exercise for migraine.

The results of this study showed that YouTube videos on exercise for migraine are generally of low quality and poor reliability. These findings contrast with the results for other YouTube videos on prescribed exercise for various health-related problems, in which most of the videos reached a high- or moderate-quality score (Kocyigit et al., 2019; Koçyiğit, Okyay & Akaltun, 2020; Culha et al., 2021; Chang & Park, 2021; Rodriguez-Rodriguez et al., 2021). However, other authors have found results similar to those of this study, with most of the videos reaching a low-quality score, although scales other than GQS and DISCERN have been used to evaluate the quality and reliability of the videos (Villafañe et al., 2018; Lee et al., 2018). The studies that found better quality scores in their included videos also found that most of the authors were healthcare professionals, which could explain the difference in scores.

Although the percentage of low-quality videos in the healthcare/exercise professional author category was higher than the percentage of high-quality videos, this category obtained more videos with higher quality scores in both GQS and DISCERN than the alternative medicine/patient author category, with statistically significant differences. These results agree with those of previous studies with higher quality scores in the videos from healthcare professionals, academics, universities and health institutions and lower quality scores in those of non-healthcare professionals, patients and independent users (Lee et al., 2018; Kocyigit et al., 2019; Koçyiğit, Okyay & Akaltun, 2020; Koçyiğit, Akyol & Şahin, 2021; Chang & Park, 2021; Rodriguez-Rodriguez et al., 2021). In the study by Culha et al. (2021), no differences were found in GQS scores between healthcare professionals/institutions and independent health information websites. However, the explanation for this outcome appears to be that most of the independent websites belonged to physical therapists, who are healthcare professionals and should therefore have been included in this category.

Regarding the video characteristics, there were also significant differences between both authors’ categories, with higher numbers of likes, dislikes, views, VR and comments in the alternative medicine/patients’ videos. Likewise, a negative correlation was found between the quality of the videos and the number of likes, dislikes, views, VR and comments. In contrast to these findings, other articles found no statistically significant differences in video characteristics regarding video sources or quality categories (Kocyigit et al., 2019; Culha et al., 2021; Koçyiğit, Akyol & Şahin, 2021; Heisinger et al., 2021; Chang & Park, 2021).

The lowest-quality migraine exercise videos appear to generate a greater impact on the population, with higher viewership and interaction rates. There were no differences between both authors’ categories in VPI, and there was no correlation between the quality scores and VPI, which could be interpreted as viewers being unable to discriminate between low- and high-quality information regarding exercise for migraine.

The high prevalence of headaches, including migraines (Vos et al., 2017), and the lack of awareness regarding prescribed exercise and its usefulness in this disorder facilitate the spread of misinformation in media resources such as YouTube, without being correctly evaluated by the users. YouTube has no filter or peer-review process for screening videos based on their source, quality or evidence (Delli et al., 2016). This context makes it possible to generate and spread misconceptions about exercise for migraine, as observed in this article, which could result in patients performing interventions that could fail to improve their condition or even worsen it.

There were no differences between the two categories regarding time the videos have been online and their duration. These characteristics could therefore not affect the differences in the other features.

In terms of grades of recommendation, aerobic exercise obtained a B grade for migraine treatment. Moderate-intensity aerobic exercise appears to be the most studied modality of prescribed aerobic exercise for migraine and also obtained a B grade, followed by high-intensity aerobic exercise and low-to-moderate continuous aerobic exercise, both obtaining a C grade. Aerobic exercise can improve migraine in various ways: it can activate anti-inflammatory systems that improve the activity of neuropeptides and other inflammatory mediators involved in triggering the trigeminal neurovascular system (Hasbak et al., 2002; Bigal et al., 2007; Beavers et al., 2010); aerobic exercise can activate central mechanisms of pain modulation, such as the descendent pain inhibitory system and endocannabinoid system (Raichlen et al., 2012; Rice et al., 2019); and aerobic exercise can alter psychological and social dimensions, improving outcome expectancies, self-efficacy and self-management strategies and reducing anxiety and depression symptoms (Salmon, 2001; Trivedi et al., 2011; Bromberg et al., 2012; Amin et al., 2018).

Yoga also obtained a B grade. The mechanisms by which yoga ameliorates migraine symptoms could be related to improved physical status and its psychological effects in reducing stress and anxiety, both of which are involved in the triggering and poorer prognosis of migraine (Wren et al., 2011; Probyn et al., 2017). Similar effects on ameliorating migraine symptoms by reducing stress could be achieved with relaxation and breathing exercises (Kaushik et al., 2005), which obtained a C grade.

In terms of lifestyle/behavior changes regarding exercise, which obtained a B grade, the introduction of better mealtime patterns or diet interventions, sleep behaviors, correct hydration and regular physical activity could improve migraine symptoms. The alteration or disruption of these patterns is related to the triggering and worsening of migraine, and the implementation of regular schedules and healthy behaviors could reduce the symptoms and impact of migraine (Khan et al., 2021).

Resistance training, which obtained a C grade of recommendation in favor of the intervention, achieves improvements in other chronic conditions, such as fibromyalgia and low back pain (Larsson et al., 2015; Wewege, Booth & Parmenter, 2018). Activating central pain modulation mechanisms and the improvement in psychological dimensions could explain the efficacy of this intervention in these conditions (Wewege, Booth & Parmenter, 2018; Rice et al., 2019). However, more research regarding the effects of resistance training for migraine treatment is needed to reduce the uncertain evidence in these patients. Moreover, strengthening of the neck muscles appears to be ineffective for reducing migraine pain, obtaining a C grade of recommendation against this intervention (Benatto et al., 2022).

Qigong and Tai Chi could improve migraine symptoms, with a D grade of recommendation. These interventions are based on case series and narrative reviews, and better study designs need to be developed to improve the conclusions as to their efficacy (Elinoff et al., 2009; Song & Chu, 2021).

Evaluating these results for grades of recommendation, almost 71% of the 90 videos included in the SIGN evaluation obtained a B grade of recommendation for their interventions. These videos were almost exclusively about aerobic exercise, yoga and lifestyle/behavior changes regarding exercise. However, most of the videos had low quality and low reliability regarding the flow and information offered. Moreover, there were no differences between both author categories’ videos regarding grades of recommendation. Healthcare and exercise professionals do not seem to have shared better evidence-based information than alternative medicine therapists and patients based on these results. However, most of grade B videos from alternative medicine/patient authors were about yoga. On the one hand, this intervention has achieved a high grade of recommendation, but most academics and healthcare professionals seem to have some negative bias toward yoga. Its eastern origin and the mystic aura that defines part of this intervention could have led to its rejection by most of the scientific community. On the other hand, the introduction of some mystical and pseudo-scientific concepts in the yoga videos evaluated in the present study has influenced the low scores obtained on the quality scales by these videos.

Thirty percent of the videos showed interventions with a C and D grade of recommendation, such as relaxation and breathing exercises, resistance exercises, Qigong and Tai Chi. Neck strengthening was shown as an effective intervention for migraine treatment in 2% of the videos included in the evaluation. However, the current classification into grades of recommendation has shown that neck strengthening is not an effective intervention for migraine pain improvement, with a C grade of recommendation. Only one study evaluated the efficacy of neck strengthening in migraine treatment (Benatto et al., 2022). If research in this field progresses, the grade of recommendation could change in the future.

The videos not included in the evaluation with grades of recommendation showed interventions that had been discarded from the classification due to lack of evidence, such as self-massage exercise and stretching of the cervical spine region. These interventions lacked evidence for reducing migraine pain and should not be presented as effective interventions on YouTube.

Limitations

YouTube has a dynamic structure that changes over time, with variations in views, likes, dislikes, comments, the release of new videos, and the deletion of older videos. This study analyzed YouTube content regarding exercise for migraine at a single time-point, and these statistics could change in the future.

Moreover, the reproducibility of the search strategy is poor due to the influence of previous searches performed by users and the annual change in the YouTube search algorithm. In this study, we used an incognito tab to suppress any influence of previous searches, but this cannot reproduce the results of other users.

The search strategy was conducted only in English and Spanish. A search in other languages should be performed to contrast the quality of videos about exercise for migraine with a better scope.

A Chrome browser extension to reveal the number of dislikes was necessary due to the recent YouTube policy on concealment of dislikes. However, this application conducts an approach of the data that depends on the number of users utilizing it to improve its accuracy.

Lastly, the quality assessment was conducted with GQS and DISCERN, scales that have a number of subjective items. However, the agreement between the reviewers was high in this study. Moreover, these scales do not seem to be able to assess the attractiveness of YouTube videos. The quality achieved by the included videos could not be related to the viewers’ preferences.

Conclusions

Prescribed exercises based on aerobic exercise, yoga and lifestyle/behavior changes regarding exercise are recommended, with a grade B according to the results of our classification into grades of recommendation. Nearly 50% of the total videos included in this study obtained this grade of recommendation, although most of them were of low quality and low reliability regarding information on the effects of exercise for migraine based on the quality evaluation. Therefore, better quality videos regarding exercise for migraine should be shared on YouTube, with better scoping strategies by healthcare and exercise professionals and with more rigorous YouTube policies for uploading videos. Moreover, better evidence-based information should be disseminated to the population to increase the quality of the content on social media.

Clinical implications

The results of this study show the need for better evidence-based information sharing on YouTube regarding the best exercise modalities and prescription parameters for migraine. Healthcare and exercise professional authors should adopt better strategies to achieve higher viewership and interaction rates, and YouTube should regulate the dissemination of videos with therapies that lack evidence for effective migraine treatment. The improvement in disclosing health information through social media is an important objective in the healthcare field considering the current relevance of the Internet in the diffusion of information to the public.

In addition, better dissemination of scientific current evidence through clinicians and the overall population would improve the knowledge regarding the best exercise interventions for migraine. The need to develop a systematic review and grades of recommendation to assess the videos’ content highlights the lack of information about exercise prescription in migraine patients among the population.

Supplemental Information

Supplemental Information 1 Descriptive data of the videos.

The data regarding Youtube videos included in the study that were analyzed for descriptive data and for inferential analysis.

Click here for additional data file.

Supplemental Information 2 GQS, DISCERN and Grades of Recommendation.

The different grades of recommendation regarding exercise modalities for migraine assigned to each video depending on the exercise modality that they show.

Click here for additional data file.

Supplemental Information 3 Percentage of exercise videos regarding Grades of Recommendation.

Click here for additional data file.

Supplemental Information 4 Codebook.

Click here for additional data file.

Additional Information and Declarations

Competing Interests

Author Contributions

Data Availability

Roy La Touche is an Academic Editor for PeerJ.

Álvaro Reina-Varona conceived and designed the experiments, analyzed the data, prepared figures and/or tables, authored or reviewed drafts of the article, and approved the final draft.

Borja Rodríguez de Rivera-Romero performed the experiments, prepared figures and/or tables, and approved the final draft.

Carlos Donato Cabrera-López performed the experiments, prepared figures and/or tables, and approved the final draft.

José Fierro-Marrero performed the experiments, prepared figures and/or tables, and approved the final draft.

Irene Sánchez-Ruiz performed the experiments, prepared figures and/or tables, and approved the final draft.

Roy La Touche conceived and designed the experiments, analyzed the data, prepared figures and/or tables, authored or reviewed drafts of the article, and approved the final draft.

The following information was supplied regarding data availability:

The raw data is available in the Supplemental Files.

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
