# Peer review of "Exercise interventions in migraine patients: a YouTube content analysis study based on grades of recommendation"

_PeerJ, doi:10.7717/peerj.14150_

## Round 0.1 · original submission · Minor Revisions

The three reviewers and I all are impressed with many aspects of your study and manuscript. Please look over the comments of the three reviewers before resubmitting a revised version of your manuscript

Reviewer 1 ·

Basic reporting

The article is up-to-date and comprehensive. Guidance for further work to be done. Well written.

The summary should reflect the findings of the article, in this sense, the findings section should be written in detail in the summary.

Experimental design

Research question well defined,Methods described with sufficient detail

Validity of the findings

Conclusions are well stated, linked to original research question

Additional comments

The article is up-to-date and comprehensive. Guidance for further work to be done. Well written, but I have a few suggestions.
In the summary, it should be written how many videos were evaluated in the method section.
Line 54-57 This definition fits better with the definition of migraine with aura
It may be more appropriate to define migraine in general here. Because the most common type of migraine is without aura (75% of cases). Migraine is a genetically influenced complex disorder characterized by episodes of moderate-to-severe headache, most often unilateral and generally associated with nausea and increased sensitivity to light and sound.
Line 122-125 Are the names of the people/organizations that prepared the videos kept confidential? You should write a sentence about it.
Line 129-138 Information should be given about the number of videos reviewed. What are the areas of expertise of the two independent reviewers?
Line 143-145 did not have sounds were excluded from the study
Line 537- 545 This section can be postponed after the conclusion
Line 586 Reference writing should be reviewed. In some references, all the first letters in the article name are capitalized. In some references, only the first letter of the first word is capitalized.

·

Basic reporting

Congratulations on the manuscript.
The study is well written and described. It presents a good literary foundation and achieves the proposed objectives. In addition, the references are recent and updated.

Experimental design

The manuscript is original and has a suitable design. The tools used are valid and reproducible. Furthermore, the description of the methods is quite complete and clear.

Validity of the findings

Lines 54-57: It would be interesting to refer to the International Classification of Headache Disorders.

Lines 68-72: No more recent references?

The conclusion is long. In this topic, the exciting thing is to bring summarized and direct information about the study's conclusion.

Reviewer 3 ·

Basic reporting

Considering the aspect of basic reporting, I suggest the authors to revise the terms used to dichotomize the authors of the analyzed videos. I think it is inappropriate to name a category as “non-profesional” authors, considering that some professionals are included at it (lines 170-175). Alternative medicine therapists and non-healthcare professionals are professionals, but they have a different profession than healthcare and exercise professionals. The justification of this choice appears later at the lines 346-349. But, still, I recommend considering another label.
The references of the coding analysis used at this first part of the study is missing (lines 161-163).
The way the objectives were presented could be revised (and consequently methods and results). It does not represent an appropriate “unit of publication”. First you search and code the videos, then you make a systematic review about the theme and finally you describe the proportion of the videos that presented the levels od evidence. It seems to me that presenting first the systematic review would be more appropriate.

Experimental design

The abstract´s objective is “This qualitative content analysis study aims to review and evaluate YouTube videos regarding exercise for migraine.” However, by applying structured scales to assess the quality of the videos and data visualization; you mostly used a quantitative design for that. The only qualitative part of the manuscript seems to be the coding process, which would need further explanation, to fit in a good quality qualitative study report (theoretical framework or base to perform this inductive analysis).
Moreover, there should be a rationale behind the coding process (lines 161-168): why do you need to codify the videos based on countries of origin? Do you have any objective related to that? It should be explained.

The third and the fourth paragraphs of your introduction give the impression that exercises are an alternative choice to the pharmacological treatment of migraine patients. Is it correct? Or do you think that both interventions, pharmacological and non-pharmacological strategies should be considered together?

Validity of the findings

My biggest concern about the discussion is that none of both categories (“professionals and non-professionals”) exceeded 10% of at least good quality at the GQS scores neither “high” classification at DISCERN scores. The discussion and the conclusion seem to advocate in favor to professionals’ videos but the results does not support it.
Another important aspect is that the correlations between You tube related metrics and the GQS and DISCERN scores do not support any solid interpretation (as for example both likes and dislikes presented the negative correlation), although several were statistical significant. I miss a critical discussion about the impossibility to associate the quality of the video as they were assessed with good metrics of impact only. Maybe the quality of videos that you are expecting has nothing to do with its attractiveness in social media/internet. It also would be of benefit to present the tools limitations.
Regarding the data presented at Figure 2, it would be also interesting to present the data about the subgroups in an additional figure. We could observe if being “professional” is somehow associated to the dissemination of scientific-based information or not.
In the way data is presented, it would be possible to interpret that although videos do not present the quality expect and assessed by the scales you applied, most of them indeed present data that are in line with a grade B of recommendation. Which is a good result about this topic, not reflected in the conclusions.

Finally, authors could admit that if there was the need to perform at the same study the systematic review about the grade of recommendation to assess if the videos content were adequate, then, those people who prepared those videos did not have this information of evidence elsewhere before it. Maybe, better dissemination of scientific findings through overall population is the main reason for a potential lack of quality.

---

## Round 0.2 · accepted · Accept

I congratulate the authors on their manuscript and their attention to detail in attending to the three reviewers' initial constructive criticisms.

Reviewer 1 ·

Basic reporting

The article is clear, litrature references are enough.

Experimental design

Methods described with sufficent detail

Validity of the findings

Conclusion are well stated.

Additional comments

Dear editor, the corrections I have stated have been made and I accept the publication of the article.
Best regards.

Reviewer 3 ·

Basic reporting

Authors answered all the questions raised and changed the manuscript properly.

Experimental design

Authors answered all the questions raised and changed the manuscript properly.

Validity of the findings

Authors answered all the questions raised and changed the manuscript properly.